# Human Serum, Following Absorption of Fish Cartilage Hydrolysate, Promotes Dermal Fibroblast Healing through Anti-Inflammatory and Immunomodulatory Proteins

**DOI:** 10.3390/biomedicines12092132

**Published:** 2024-09-19

**Authors:** Julie Le Faouder, Aurélie Guého, Régis Lavigne, Fabien Wauquier, Line Boutin-Wittrant, Elodie Bouvret, Emmanuelle Com, Yohann Wittrant, Charles Pineau

**Affiliations:** 1Abyss Ingredients, 860 Route de Caudan, 56850 Caudan, France; elodie@abyss-ingredients.com; 2Univ Rennes, CNRS, Inserm, Biosit UAR 3480 US_S 018, Protim core facility, F-35000 Rennes, France; aurelie.gueho@univ-rennes.fr (A.G.); regis.lavigne@univ-rennes.fr (R.L.); , charles.pineau@univ-rennes.fr (C.P.); 3Univ Rennes, Inserm, EHESP, Irset (Institut de Recherche en Santé, Environnement et Travail) - UMR_S 1085, F-35000 Rennes, France; 4Clinic’n’Cell SAS, Faculty of Medicine and Pharmacy, TSA 50400, 28 Place Henri Dunant, 63001 Clermont-Ferrand, France; fabien.wauquier@clinicncell.com (F.W.); yohann.wittrant@inrae.fr (Y.W.); 5UNH, UMR1019, INRAE, 63009 Clermont-Ferrand, France; 6Human Nutrition Unit, Clermont Auvergne University, BP 10448, 63000 Clermont-Ferrand, France

**Keywords:** skin health, marine collagen peptides, glycosaminoglycans, proteomics, wound healing, primary cells, human dermal fibroblasts

## Abstract

**Background/Objectives**: Marine collagen peptides (MCPs) and glycosaminoglycans (GAGs) have been described as potential wound-healing (WH) agents. Fish cartilage hydrolysate (FCH) is a natural active food ingredient obtained from enzymatic hydrolysis which combines MCPs and GAGs. Recently, the clinical benefits of FCH supplementation for the skin, as well as its mode of action, have been demonstrated. Some of the highlighted mechanisms are common to the WH process. The aim of the study is therefore to investigate the influence of FCH supplementation on the skin healing processes and the underlying mechanisms. **Methods**: To this end, an ex vivo clinical approach, which takes into account the clinical digestive course of nutrients, coupled with primary cell culture on human dermal fibroblasts (HDFs) and ultra-deep proteomic analysis, was performed. The effects of human serum enriched in circulating metabolites resulting from FCH ingestion (FCH-enriched serum) were assessed on HDF WH via an in vitro scratch wound assay and on the HDF proteome via diaPASEF (Data Independent Acquisition—Parallel Accumulation Serial Fragmentation) proteomic analysis. **Results**: Results showed that FCH-enriched human serum accelerated wound closure. In support, proteins with anti-inflammatory and immunomodulatory properties and proteins prone to promote hydration and ECM stability showed increased expression in HDFs after exposure to FCH-enriched serum. **Conclusions**: Taken together, these data provide valuable new insights into the mechanisms that may contribute to FCH’s beneficial impact on human skin functionality by supporting WH. Further studies are needed to reinforce these preliminary data and investigate the anti-inflammatory and immunomodulatory properties of FCH.

## 1. Introduction

The epidermis of human skin is essential for tissue homeostasis, constituting the most important innate defense barrier against all pathogens [1]. Skin lesions can be caused by genetic disorders, and other diseases or by common interactions with the environment (e.g., dryness, scarring, infections, and burns) [2,3]. Restoring skin integrity involves multidimensional processes such as inflammation, proliferation, epithelialization, angiogenesis, remodeling, and healing [4]. Several advanced medical devices, such as hydrogels or nanofibrous scaffolds, have been developed for the management of wound healing (WH). Most of them include a wide range of products containing synthetic and natural (e.g., collagen, hyaluronic acid and chitosan) polymers, as well as bioactive molecules, and possess antimicrobial, anti-inflammatory and antioxidant properties [5].

Marine collagen peptides (MCPs) have been shown to be an effective biomaterial for WH and skin regeneration [6]. Produced from collagen by chemical or enzymatic hydrolysis, their low molecular weight increases their solubility in water, making them more easily assimilated and absorbed [7]. Hu et al. demonstrated that the application of MCPs isolated from the skin of tilapia improved wound closure in an in vitro scratch assay and in experiments on deep partial-thickness scald wounds in rabbits [8]. Several groups have shown that oral administration of MCPs isolated from fish had beneficial effects on skin healing in wounded rats. Compared to control groups, recovery rates, hydroxyproline (amino acid frequently present in collagen) content, epithelialization, fibroblast proliferation, vascularization as well as angiogenesis were increased in the treated groups [9,10,11].

Glycosaminoglycans (GAGs) from marine sources, essential components of bone and cartilage tissues, have been described as potential anti-inflammatory agents. They are linear polysaccharide chains composed of repeating disaccharide units, which can be differently sulphated, and are generally grouped into four groups: hyaluronic acid (HA) or hyaluronan; keratan sulfate; heparan sulfate/heparin; and chondroitin sulfate (CS)/dermatan sulfate (DS) [12]. The addition of CS isolated from shark cartilage was shown to have significant anti-inflammatory activity in vitro. It allowed the reduction of inflammation mediators and apoptosis in mouse articular chondrocytes after stimulation with lipopolysaccharides (LPS) [13] and produced a decrease in both nitric oxide (NO) and pro-inflammatory cytokines release and an increase of the anti-inflammatory cytokine interleukin-10 in mouse bone-marrow derived macrophages stimulated with LPS and interferon-γ [14]. Marine GAGs were also shown to impact wound repair. Krichen et al. showed that the application of GAGs from fish skins-based gels, on dermal full-thickness excision wounds in a mouse model, enhanced significantly WH activity. The treatment reduced the edema after carrageenan injection, protected edema tissue from oxidative damage, and reduced the risk of inflammation [15].

Fish cartilage hydrolysate (FCH) is a natural active food ingredient obtained from enzymatic hydrolysis which combines collagen peptides and GAGs. Previously, FCH was shown to have a significant anti-inflammatory activity on primary human articular chondrocytes [16]. In a randomized, double-blind, placebo-controlled clinical trial, the clinical benefits of FCH supplementation for the skin were demonstrated, with significant reduction in wrinkles and increased dermal density in healthy middle-aged women [17]. Recently, using an ex vivo clinical approach, the bioavailability and safety of FCH in humans, as well as its nutricosmetic potential, were demonstrated. FCH was shown to promote hydration, elasticity, and to limit the expression of catabolic factors involved in ECM degradation. Indeed, FCH stimulated the growth of human dermal fibroblasts (HDFs), HA production, elastin synthesis, and TGF-β release, while inhibiting the expression of matrix metalloproteinases (MMP)-1 and 3 [18].

Collagenolysis and elastolysis by MMPs occur in WH [19]. Likewise, TGF-β is involved in fibroblast proliferation and keratinocyte differentiation in WH [20,21]. As mentioned above, MCPs and marine GAGs, to a lesser extent, are studied for their facilitating role in the healing process. Thus, supplementation with FCH, which combines MCPs and GAGs, could have a beneficial effect on wound healing. The aim of the study is therefore to investigate the influence of FCH on skin healing processes and the underlying mechanisms. To this end, an ex vivo clinical approach, which considers the clinical digestive course of nutrients, coupled with primary cell culture on human dermal fibroblasts (HDFs) and ultra-deep proteomic analysis, was performed. The effects of human serum enriched in circulating metabolites resulting from FCH ingestion (FCH-enriched serum) were first assessed on HDF WH via an in vitro scratch wound assay. Next, its impact on the HDF proteome was explored using a diaPASEF (Data Independent Acquisition—Parallel Accumulation Serial Fragmentation) proteomic analysis. By accelerating scratch wound healing of HDFs and stimulating the expression of proteins with anti-inflammatory and immunomodulatory properties or prone to promote hydration and ECM stability, FCH could contribute to skin functionality by supporting WH.

## 2. Materials and Methods

### 2.1. Product Tested

Fish cartilage hydrolysate (FCH) is a water-soluble powder obtained from a standardized manufacturing process based on enzymatic hydrolysis of marine fish cartilage, without preservatives or processing aids (Abyss Ingredients, Caudan, France). Its natural composition, containing 65% of marine collagen peptides (MCPs) and 25% of chondroitin sulfate (CS) has been determined by Dumas and Scott assays, respectively. The molecular weight distribution of MCPs was determined by gel permeation HPLC-UV: more than 98% has a molecular weight under 3000 Da and 80% under 1000 Da (Appendix A). These analyses were carried out by accredited laboratories (COFRAC). FCH was given at a dose of 12 g per intake and per volunteer. A single dose corresponded to an acute exposure to 8.04 g of MCPs and 3.24 g of CS. Animal toxicity studies have demonstrated the absence of deleterious effects for these compounds. For CS, a single dose of 2 g/kg was tested on rodents without any adverse effects [22]. The LD50 for oral ingestion of HC in rodents is over 10 g/kg [23].

### 2.2. Human Study Protocol

The human ex vivo methodology used in this study has been registered as a written invention disclosure by the French National Institute for Agronomic, Food and Environment Research (INRAE). The number of volunteers (*n* = 10) was calculated to provide enriched serum for the cell culture studies that form the core of the ex vivo demonstration, rather than to support benefits. This number proved sufficient to demonstrate nutraceutical benefits and mechanisms [24,25,26,27,28].

Ten healthy men (age: 25.4 ± 3.7 years old,; BMI: 23.6 ± 1.9 kg/m^2^,; >60 kg; drug-free; and regardless of ethnicity) volunteered for this study. They were checked for normal blood formulation, renal function (urea and creatinine), liver function (aspartate aminotransferase, alanine aminotransferase and gamma-glutamyltransferase activity), and the absence of allergy to the product. After fasting for 12 h, they were given 12 g of FCH and had their blood drawn. A total of 48 mL of venous blood was drawn before ingestion for the naive serum sample (H-NAIVE) and at FCH absorption peak (140 min after ingestion) for the FCH-enriched human serum sample (H-FCH). It was previously shown that following ingestion of FCH, the circulating concentration of hydroxyproline continuously increases to reach a maximum of 117.7 µM at 140 min post-ingestion (+87.5% compared to basal level), before returning to almost basal level by the end of the monitoring period. The same observation was made for chondroitin sulfate. For this reason, blood collection occurred at 140 min after ingestion [18]. Serum was stored at −80 °C until analysis. The human study protocol used in this study and described in [18] is also detailed in the Appendix A.

### 2.3. Human Primary Dermal Fibroblast (HDF) Cultures

HDFs were obtained from an adult donor (Sigma-Aldrich, Lyon, France, 106-05A). A cell culture was carried out in Dulbecco’s Modified Eagle Medium (DMEM, Biowest, Nuaillé, France, L0066-500) supplemented with 10% fetal calf serum (FCS, Invitrogen Corporation, Illkirch, France) and 1% penicillin/streptomycin (Life Technologies, Villebon-Sur-Yvette, France), at 37 °C and 5% CO_2_/95% air.

### 2.4. In Vitro Scratch Assay

#### 2.4.1. Cell Treatment

HDFs were seeded at a density of 15,000 cells/cm^2^ in 24-well plates with 500 µL of culture medium and allowed to grow until they reached 80–90% confluence in the maintenance medium. HDFs were then incubated, according to the Clinic’n’Cell protocol, in DMEM supplemented with 1% penicillin/streptomycin in the presence of 10% of H-NAIVE or H-FCH serum. It was previously shown that these sera had no cytotoxic effect on human dermal fibroblast cultures [18]. Human sera were not pooled, and each monolayer was incubated with a specific serum, corresponding to 20 biological replicates (H-NAIVE, *n* = 10; H-FCH, *n* = 10). After 24 h of pre-treatment, monolayers of dermal fibroblasts were subjected to a linear scratch in the middle of the well using a P1000-tip (Figure 1). Then, culture media were changed to removed floating cells and scratched monolayers were placed under a videomicroscope allowing timelapse investigations at 37 °C and 5%CO_2_/95% air. X, Y and Z coordinates were set for each well to follow the migration of the primary HDFs covering the scratched area over 24 h. Surface covered by cells was quantified using ImageJ software 1.53 and calculated as a percentage (ratio) of the initial wounded surface to avoid bias in comparison.

#### 2.4.2. Statistics

Statistical tests and corresponding figures were carried out with Prism V.9.4.1 (GraphPad Software, San Diego, CA, USA) according to the following statistical plan. The Shapiro–Wilk normality test was performed to assess the Gaussian distribution. A non-parametric Kruskal–Wallis test was used, followed by Dunn’s test for post hoc comparisons, in the case of non-normal distribution. For multiple comparisons, measurements were subjected to one-way analysis of variance (ANOVA) with Tukey’s test, assuming normal distribution and equal variance. Values are presented as mean ± standard deviation (SD), unless otherwise indicated. Differences were considered statistically significant at *p* < 0.05.

### 2.5. DiaPASEF Proteomic Analysis

#### 2.5.1. Cell Treatment

HDFs were seeded at a density of 15,000 cells/cm^2^ in 12-well plates with 1 mL of culture medium and grown for 3 days until they reached 70% confluence in the maintenance medium. HDFs were then incubated in DMEM supplemented with 1% penicillin/streptomycin in the presence of 10% of H-NAIVE or H-FCH serum, according to the Clinic’n’Cell protocol. Human sera were not pooled, and each monolayer was incubated with a specific serum, corresponding to 20 biological replicates (H-NAIVE, *n* = 10; H-FCH, *n* = 10). After 24 h of incubation, culture supernatants were discarded and cells were washed two times with ice-cold isotonic buffer (phosphate buffered saline, PBS). The remaining liquid was carefully removed from the well and the cell monolayer was immediately stored at −80 °C. By the time the cells were frozen, cultures approximately reached 85% confluency for about 200,000 cells/well (per tested condition).

Protein extraction was carried out using the PreOmics iST kit (PreOmics GmbH, Martinsried, Germany). Each cell sample was resuspended in 70 µL of iST Lysis buffer and reduced and alkylated for 10 min at 95 °C at 1000 rpm.

#### 2.5.2. Spectral Library Creation

For construction of a spectral library, a pool of 10 µg of proteins was prepared for each condition (H-NAIVE/H-FCH) by pooling 1 µg of each protein sample. Pools were then separated by sodium dodecyl sulfate polyacrylamide gel electrophoresis (SDS-PAGE; Nu-PAGE 4–12%, Invitrogen) for 45 min at 200 V, and the gel was further stained with ReadyBlue™ Protein Gel Stain (Sigma-Aldrich, Saint-Quentin-Fallavier, France). Each lane was cut into 12 strips which were processed for in-gel digestion as previously described [29], using 4 ng.µL^−1^ of a Tryspine/LysC mix (v5073, Promega, Madison, WI, USA) in 25 mM ammonium bicarbonate / 0,01% ProteaseMAX^TM^ (Promega, Madison, WI, USA). After desalting using the Phoenix kit (PreOmics, Martinsried, Germany), each fraction of enzymatically digested proteins (about 200 to 300 ng) was analyzed in Data-Dependent Analysis (DDA) and PASEF mode to generate the spectral library. Parameters for the nanoliquid chromatography-tandem mass spectrometry (NanoLC-MS/MS) experiment, described previously [30], are available in the Appendix A.

#### 2.5.3. Sample Preparation and Acquisition

Ten micrograms of each sample (H-NAIVE, *n* = 10; H-FCH, *n* = 10) were digested by adding 50 µL of Digestion PreOmics Mix (Trypsin and LysC) and incubation at 37 °C for 3 h. Digested samples were then desalted according to manufacturer’s instructions and dried by vacuum centrifugation. Dried peptides were resuspended in PreOmics LC-LOAD buffer at a concentration of 100 ng/µL. The twenty samples were then analyzed individually in diaPASEF mode. The parameters of the nanoLC-MS/MS experiment have been described previously [30] and are also detailed in the Appendix A.

#### 2.5.4. MS Data Processing

The ion mobility resolved mass spectra, nested ion mobility versus *m/z* distributions, and fragment ion intensity sums were extracted from the raw data file using DataAnalysis 6.0 (Bruker Daltonik GmbH, Bremen, Germany). The signal-to-noise (S/N) ratio was increased by summing the individual Trapped Ion Mobility Spectrometry (TIMS) scans. Mobility peak positions and half-peak widths were determined on the basis of extracted ion mobilograms (EIM, ±0.05 Da) using the implemented peak detection algorithm. Feature detection was carried out.

#### 2.5.5. Data Analysis—Library Generation

Raw DDA files were analyzed in Spectronaut version 16 software (Biognosys, Schlieren, Switzerland), using the integrated Pulsar search engine and a search scheme with default parameters to generate the corresponding spectral library. The calibration search was dynamic, while the MS1 and MS2 correction factors were 1. Data were searched in the UniProt KB Human database (20,594 sequences, downloaded February 2023), with trypsin/P as protease with up to two missed cleavages. Carbamidomethylation of cysteine residues was set as a fixed modification. Methionine oxidation and acetylation of protein N-termini were set as variable modifications. A false discovery rate (FDR) of less than 1% was ensured at the precursor, peptide and protein levels.

#### 2.5.6. Library Search of DIA Data

During the conversion into htrms files using the htrms converter (Biognosys, Schlieren, Switzerland), MS1 and MS2 data were centroided, while the other parameters were set to default. Using the previously generated libraries and default settings, the htrms files were then analyzed with Spectronaut. Results were filtered by a 1% FDR at the precursor, peptide, and protein levels using a target-decoy approach corresponding to a *q*-value ≤ 0.01.

#### 2.5.7. Quantification and Statistical Analyses of Proteomics Data

Quantification based on average top 3 and data normalization were performed using Spectronaut software. Given the number of samples analyzed (less than 500 individuals), the local regression normalization described by Callister et al. [31] was carried out for the whole dataset. A t-test was applied, and data were filtered using a Q-value (multiple testing corrected *p*-value) of 0.05 and an absolute log2 ratio of 0.58, which correspond to a fold change of 1.5.

The list of differential proteins between both groups (H-NAIVE, H-FCH) was classified using an unsupervised Bayesian clustering [32]. Each generated cluster was grouped according to protein expression then used for gene ontology (GO) term enrichment analysis with Cytoscape’s Bingo plug-in (Ghent, Belgium) [33]. The right-sided hypergeometric test with a Benjamini-Hochberg correction, a *p*-value threshold of 0.05, and the use of all quantified proteins as the reference set, were parameterized. Visualization of enriched GO terms was performed according to the protocol described by [34]. Principal component analysis (PCA) and volcano were produced by Spectronaut, while box plots were generated using the MetaboAnalyst 5.0 web server (Edmonton, Canada) [35].

## 3. Results

### 3.1. FCH-Enriched Serum Accelerates Scratch WH of HDFs

Primary HDFs were subjected to scratch wound assays in order to investigate the potential benefit of FCH in skin healing. As shown in Figure 2A, the HDF wound closed more rapidly in the presence of H-FCH than in the presence of H-NAIVE serum. The percentage of surface area covered by cells increased significantly at 6 h, from 9.5% with the addition of H-NAIVE to 15.5% with the addition of H-FCH. This percentage tended to increase at 12 h, in the presence of H-FCH compared with H-NAIVE (*p* = 0.06; 41.8% vs. 34.2%) (Figure 2B). Therefore, FCH-enriched serum accelerates scratch WH of HDFs.

### 3.2. FCH-Enriched Serum Increases Expression of Proteins Involved in WH

Primary HDFs were subjected to a proteomic study in order to evaluate the underlying mechanisms of FCH on skin functionality. In the overall series (*n* = 20), 6842 proteins were quantified from HDFs (Appendix A). The PCA of quantified proteins discriminated HDFs exposed to H-NAIVE from those exposed to H-FCH serum (Figure 3), supporting that HDF protein expression is modulated by FCH-enriched serum.

The T-test displayed 195 differential proteins (Q < 0.05; absolute log2 ratio of 0.58) between H-FCH and H-NAIVE, with 122 and 73 proteins over- and under expressed in H-FCH, respectively (Appendix A).

Clustering of the 195 differential proteins revealed six clusters (C1–C6) (Figure 4A, Appendix A). Two clusters brought together proteins with increased expression following exposure to H-FCH (C1, C2). Proteins upregulated in cluster C1-C2 corresponded to enrichment in pathways such as acute inflammatory response (cluster frequency 19/91 (20.8%), corrected *p* = 1.0068.10^−17^), immune response (cluster frequency 27/91 (29.6%), corrected *p* = 1.8098.10^−16^), response to wounding (cluster frequency 23/91 (25.2%), corrected *p* = 1.1189.10^−11^), and glycosaminoglycan binding (cluster frequency 5/91 (5.4%), corrected *p* = 1.8467.10^−2^) (Figure 4B, Appendix A). Some of these pathways and the associated proteins are presented in Table 1. All the proteins included in the GO term “acute inflammatory response” were common to the GO term “response to wounding”. Likewise, from the 27 proteins included in the GO term “immune response”, 10 were common to the GO term “response to wounding”.

Two clusters brought together proteins with decreased expression following the exposition to H-FCH (C3, C6) (Appendix A). Proteins from these clusters did not display pathway enrichment but matched with the terms: metabolic process (cluster frequency 28/40; with proteins such as 9S ribosomal protein L23), mitochondrial (L23mt), MRP-L23), catalytic activity (cluster frequency 20/40; e.g., Alpha-1,2-mannosyltransferase, ALG9), and organelle part (cluster frequency 20/40; e.g., Aquaporin-1, AQP-1).

Some of the differential proteins further described in the discussion are illustrated in Figure 4C,D, representing their normalized intensity and their statistical comparison between H-NAIVE and H-FCH.

## 4. Discussion

In this study, in order to investigate the potential nutraceutical benefits of FCH for human skin functionality, an ex vivo clinical approach which considered the clinical digestive course of nutrients was coupled with primary cell culture on HDFs. This approach represents a physiological and ethical alternative to preclinical studies, giving in vitro models a clinical dimension for assessing the impact of nutraceuticals [24,25,26,27,28]. The strength of this method is that it takes into account the overall effects of the product on health. These effects may result from the direct action of bioactive compounds on the final target, or indirectly via an organ or blood cells. Next, a diaPASEF proteomic analysis, enabling in-depth proteome detection [36,37], was carried out to explore the mechanism of action of FCH-enriched human serum on HDFs.

The bioavailability, safety, and the beneficial effects of FCH on skin health have been demonstrated previously, promoting the production of macromolecules present in the ECM [18]. The effects of FCH on the WH process were studied here, using an in vitro fibroblast-mimicking lesion. FCH-enriched serum led to an acceleration in scratch wound healing of HDFs. Wound closure was observed as early as 6 h of incubation, suggesting a specific enhancement of the cell migration process, rather than a simple proliferation artefact, warranting further investigation. For primary human fibroblasts, the cell cycle lasts between 16 and 28 h with a mean of 20 h [38]. To date, there was no impact of FCH-enriched serum on HDF proliferation after 24 h incubation [18]. As mentioned above, MCPs and marine GAGs, to a lesser extent, have been studied for their facilitating role in the healing process [6,8,9,10,11,15]. Indeed, they can contribute to skin tissue engineering by promoting both the proliferation and migration of primary HDFs and the differentiation and the migration of human keratinocytes [6,39]. In addition, FCH-enriched serum has previously been showed to increase the release of TGF-β [18], which is also involved in fibroblast proliferation and keratinocyte differentiation in WH [20,21]. Overall, these results suggest that FCH could be a promising nutrient for WH.

Using a proteomic strategy, it was shown that FCH-enriched serum increases the expression of proteins involved in response to wounding, in particular several members of the serpin family. Serpins (serine protease inhibitors or classified inhibitor family I_4_) are a widespread family of protease inhibitors that use conformational change to inhibit target enzymes. They play a central role in the control of many important proteolytic cascades, such as the mammalian coagulation pathway [40]. Serpins are of particular interest in WH due to their inhibitory effects on specific proteases that are relevant to wound response including inflammation, ECM remodeling, and cell migration and proliferation [41]. Alpha1-antitrypsin (SERPINA1) has been described to reduce the activity of MMP-9, as well as to have anti-inflammatory and anti-apoptotic effects during the healing response [42,43]. Likewise, serpin A1, known as a potent inhibitor of neutrophil elastase, which hydrolyzes proteins including elastin, has therapeutic potential as a wound healing agent [44]. Furthermore, alpha1-antichymotrypsin (SERPINA3) showed similar effects to alpha1-antitrypsin during WH, accelerating wound closure in an experimental open skin wounds in rabbits [45]. Finally, local administration of plasma protease C1 inhibitor (SERPING1) has been shown to inhibit edema formation, reduce inflammatory tissue damage, and increase re-epithelialization of cutaneous burn lesions in animals [46,47]. FCH-enriched serum has previously been demonstrated to inhibit MMP-1 and 3 expression while increasing elastin production in HDFs [18]. This increase was confirmed here by a proteomic approach (elastin: Q-value = 1.94 × 10^−6^; absolute log2 ratio = 0.52) (Appendix A).

FCH-enriched serum also induced the upregulation of several members of the complement activation pathway. Best known for its role in immune surveillance and inflammation, there is increasing evidence that complement activation contributes to tissue repair [48]. Thus, increased collagen/fibronectin levels and enhanced WH were seen after topical application of complement C3 and C5 to rat skin wounds [49,50]. Other proteins from the response to wounding pathway, such as serotransferrin (TF), alpha-2-macroglobulin (A2M), or alpha1-acid glycoprotein (ORM) 1 and 2 were also upregulated by FCH-enriched serum. Indeed, TF was considered a potential wound healing mediator in human nasal fibroblast conditioned medium [51]. The different effects (e.g., stimulation of cell proliferation and migration, interaction with collagen) of A2M and ORM, proteins with anti-inflammatory and immunomodulating properties, may suggest a beneficial role in WH [52,53].

As aforementioned for several members of the complement activation pathway, FCH-enriched serum upregulated proteins involved in the immune response. This also includes immunoglobulins. Indeed, constant region heavy chains of immunoglobulin A (1,2), G (1–4) and M, as well as constant and variable domain of immunoglobulin light chains, were increased. Nishio and collaborators have demonstrated that B cells, which produce antibodies against damaged tissues, are involved in the process of WH. Indeed, they showed that splenectomy delayed WH and that the transfer of spleen cells into splenectomized mice recovered the WH delay. Additionally, these authors showed that immunoglobulin G1 (IgG1) bound to injured tissue, and that splenectomy reduced the amount of IgG1 binding to injured tissues [54].

Additionally, FCH-enriched serum induced the upregulation of other proteins involved in the immune response that have been described for their beneficial roles in skin pathologies such as atopic dermatitis (AD) and psoriasis (e.g., zinc-alpha-2-glycoprotein (AZGP1, ZAG), guanylate-binding protein 2 (GBP2) and N-acetylmuramoyl-L-alanine amidase (PGLYRP2)). AD and psoriasis are major chronic inflammatory skin diseases. AD is caused by a complex interplay between the immune system, skin barrier abnormalities and deregulation of the cutaneous microbiome [55,56]. Psoriasis is caused by multigenic predisposition, environmental factors, and aberrant immune response, and can alter the skin, nails, and joints [57]. Noh et al. reported that zinc-alpha-2-glycoprotein plays a role in AD pathogenesis and demonstrated that topical treatment with this protein restored the integrity of the skin barrier and limited AD inflammation [58]. The guanylate-binding protein 2 is a member of the GBP family, which is primarily known for its action against invading microbes and pathogens, as part of the innate immune response [59]. Bowcock et al. found GBP1 and 2 transcripts are differentially expressed in involved psoriatic skin versus normal skin [60]. Peptidoglycan Recognition Proteins (PGRPs or Pglyrps) represent a class of innate immunity proteins expressed in the skin. Park et al. demonstrated that Pglyrp2 protected mice from psoriasis-like skin inflammation by promoting Treg and limiting Th17 responses [61]. Overall, these results confirm the healing promotion properties of FCH-enriched serum and suggest the induction of potential anti-inflammatory and immunomodulatory properties that should be further investigated in skin pathologies.

Finally, FCH-enriched serum upregulated proteins related to GAG binding such as CCN family member 1 (CCN1), an heparin-binding, extracellular matrix-associated protein which appears to play a role in WH by up-regulating the expression of a number of genes involved in angiogenesis, inflammation and matrix remodeling (e.g., MMP1 and MMP3) in skin fibroblasts [62]. FCH-enriched serum also induced the upregulation of polypeptides involved in GAG metabolic process such as inter-alpha-trypsin inhibitor heavy chain H1 and 2 (ITIH1, ITIH2). Recent in vivo and in vitro studies have shown that HC1 and HC2 are linked to HA, resulting in the improvement of ECM stability [63]. Furthermore, it was previously demonstrated that HA synthesis was significantly stimulated in the presence of FCH-enriched serum [18]. Aquaporin-1 (AQP-1) is one of the proteins down-regulated by FCH-enriched serum. By controlling cell and tissue water content, it has been described to act in particular on inflammation, angiogenesis, and ECM remodeling. AQP1 has been shown to increase in dermal fibroblasts from patients with systemic sclerosis, probably contributing to tissue fibrosis and edema [64]. Thus, FCH could contribute to skin functionality by promoting hydration and ECM stability.

This study had several limitations. The HDF proteome was studied, comparing the influence of FCH-enriched human serum with that of naive human serum. Firstly, despite all the precautions taken to thoroughly wash the cells and remove the supernatant prior to proteomic study on HDFs, it cannot be ruled out that some of the differential proteins observed originate from serum. Indeed, immunoglobulins are produced by B cells rather than by HDFs; therefore, the observed difference in this class of protein more likely resulted from immunoglobulins attached to HDFs. In this light, one may also question the difference regarding the proteins of the complement complex. In this case, fibroblasts from the dental pulp have been reported to express all the proteins required for efficient complement activation, including C5b-9 and C5a [65]. More specifically, at the skin level, HDFs were previously demonstrated to successfully produce complement protein C4 [66], C1r and C1s [67]. Thus, even if bias may occur for certain class of proteins, a differential expression remains between the two conditions and clearly indicates a potent and positive impact of FCH on HDFs metabolism. On the other hand, the proteomic analysis was carried out in the absence of scratching, in order to avoid biases related to fibroblast injury. Thus, these preliminary results would need to be validated following scratching or in a context of IL-1β-induced inflammation.

## 5. Conclusions

The clinical benefits of fish cartilage hydrolysate (FCH) supplementation for the skin health, and its mode of action, have recently been demonstrated. In this study, using an ex vivo clinical approach coupled with primary cell culture on HDFs and diaPASEF proteomic analysis, the beneficial effects of FCH in the healing process and its potential underlying mechanisms were highlighted. Results showed that FCH-enriched human serum accelerated wound closure in an in vitro scratch assay. Accordingly, proteins with anti-inflammatory and immunomodulatory properties and proteins prone to promote hydration and ECM stability showed increased expression in HDFs after exposure to FCH-enriched serum. Taken together, these data provide valuable new insights into the mechanisms that may contribute to FCH’s beneficial impact on human skin functionality by supporting WH. The synergistic action of FCH supplementation combined with available biomedical devices for wound closure could provide an innovative strategy for the management of WH and skin generation. Further studies are needed to reinforce these preliminary data. In addition, the potential benefits of FCH supplementation in skin diseases, such as AD and psoriasis, which are closely associated with alterations in the immune system, could be investigated. To this end, the anti-inflammatory and immunomodulatory properties of FCH could be explored in more physiological preclinical models, such as skin organoids or skin explants, prior to a possible clinical trial.

## 6. Patents

The human ex vivo methodology used in this study has been registered as a written invention disclosure by the French National Institute for Agronomic, Food and Environment Research (INRAE) (DIRV#18-0058). Clinic’n’Cell^®^ has been registered as a trademark.

## Figures and Tables

**Figure 1 biomedicines-12-02132-f001:**
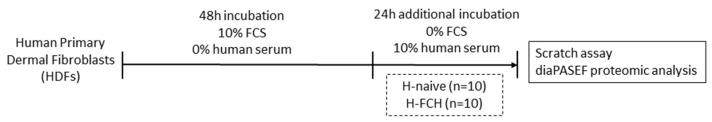
Experimental protocol for ex vivo clinical approach coupled with in vitro HDF experiments.

**Figure 2 biomedicines-12-02132-f002:**
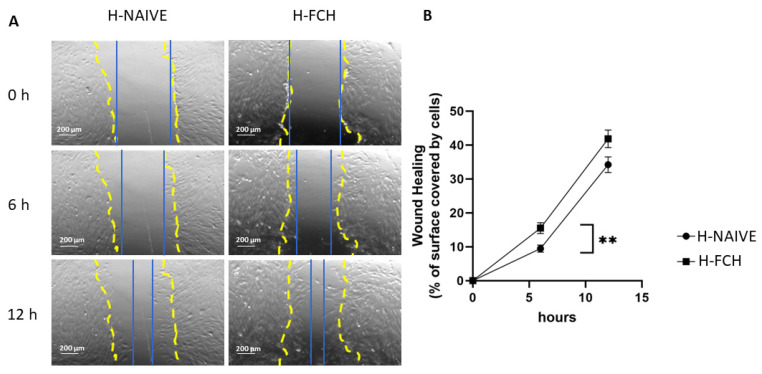
Wound-filling analysis of HDFs during a scratch wound assay. (**A**): Representative view at 0, 6, and 12 h after exposure to H-NAIVE or H-FCH. The initial scratch mask is shown in dotted yellow, while the evolution of cell migration is shown in solid blue; scale bar = 200 µm. (**B**): Percentage of covered surface by cell over time (H-NAIVE, *n* = 10; H-FCH, *n* = 10). Values are presented as mean ± SD unless specified otherwise. Differences were considered statistically significant at *p* < 0.05 with ** for *p* < 0.01.

**Figure 3 biomedicines-12-02132-f003:**
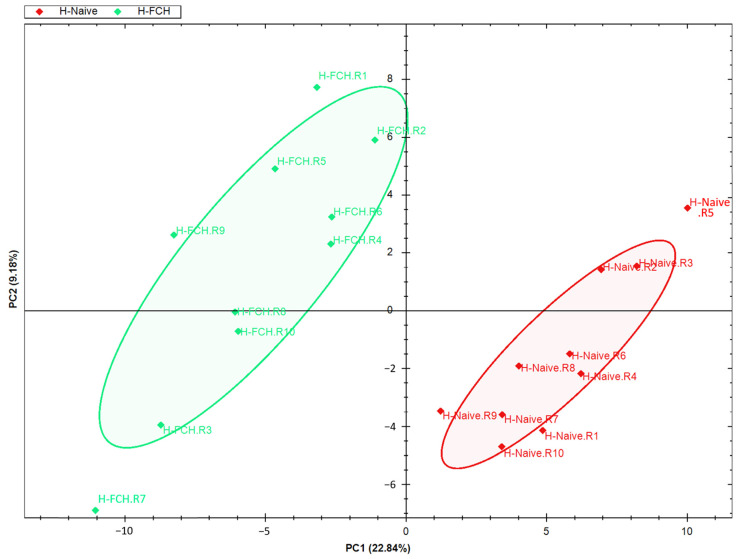
Evaluation of quantified HDF proteins. Graphic representation defined by the first two principal components (PC1 and 2) of the principal component analysis (PCA).

**Figure 4 biomedicines-12-02132-f004:**
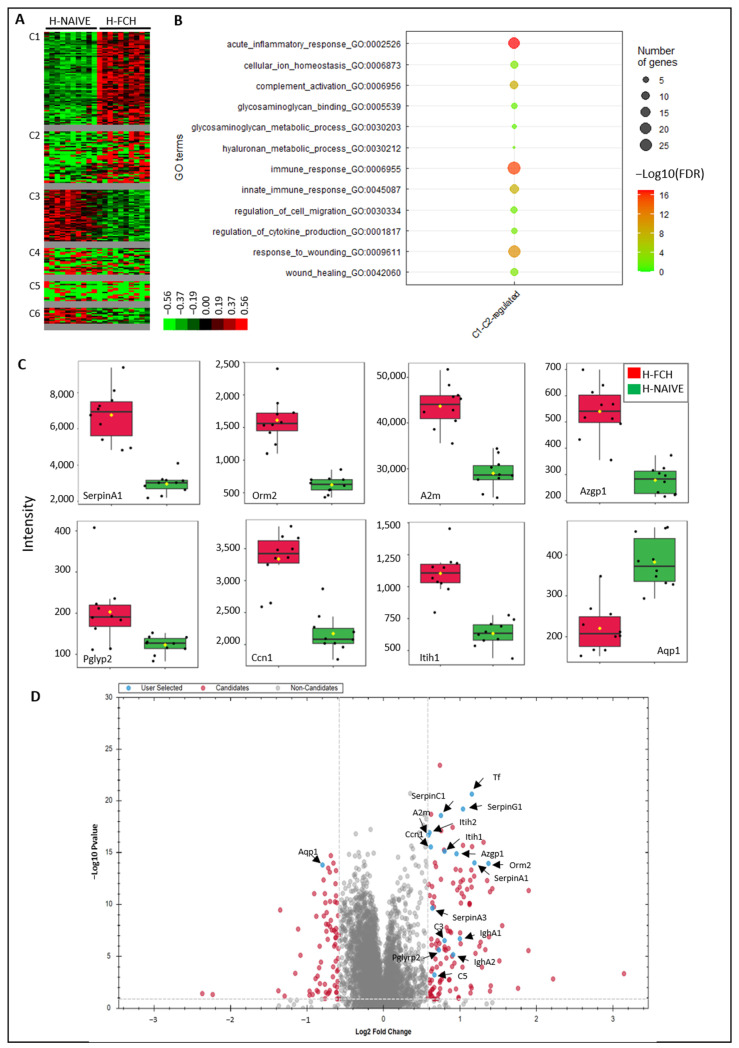
Proteins differentially expressed between H-NAIVE and H-FCH (*n* = 195). (**A**): Unsupervised Bayesian clustering (*n* = 10/group). (**B**): Gene ontology (GO) term enrichment for proteins upregulated by H-FCH serum from cluster C1-C2. (**C**): Boxplots representing normalized abundances of differentially expressed proteins. (**D**): Volcano plot representing the statistical comparison of the protein intensities of H-FCH versus H-NAIVE. The abscissa reports the fold change in logarithmic scale (difference), and the ordinate −log(Q) of *t*-tests. Some deregulated proteins involved in healing are annotated and highlighted in blue.

**Table 1 biomedicines-12-02132-t001:** Pathways and their associated proteins upregulated by H-FCH serum.

Pathway	Protein IDs	Gene Name	Protein Name
Response to wounding	P01023	A2M	Alpha-2-macroglobulin
	P02743	APCS	Serum amyloid P-component
	P02652	APOA2	Apolipoprotein A-II
	P02749	APOH	Beta-2-glycoprotein 1
	P02746	C1QB	Complement C1q subcomponent subunit B
	P02747	C1QC	Complement C1q subcomponent subunit C
	P09871	C1S	Complement C1s subcomponent
	P06681	C2	Complement C2
	P01024	C3	Complement C3
	P01031	C5	Complement C5
	P02748	C9	Complement component C9
	P21926	CD9	CD9 antigen
	P00751	CFB	Complement factor B
	P08603	CFH	Complement factor H
	P00742	F10	Coagulation factor X
	P02763	ORM1	Alpha-1-acid glycoprotein 1
	P19652	ORM2	Alpha-1-acid glycoprotein 2
	P35542	SAA4	Serum amyloid A-4 protein
	P01009	SERPINA1	Alpha-1-antitrypsin
	P01011	SERPINA3	Alpha-1-antichymotrypsin
	P01008	SERPINC1	Antithrombin-III
	P05155	SERPING1	Plasma protease C1 inhibitor
	P02787	TF	Serotransferrin
Immune response	P25311	AZGP1	Zinc-alpha-2-glycoprotein
	P02746	C1QB	Complement C1q subcomponent subunit B
	P02747	C1QC	Complement C1q subcomponent subunit C
	P09871	C1S	Complement C1s subcomponent
	P06681	C2	Complement C2
	P01024	C3	Complement C3
	P01031	C5	Complement C5
	P02748	C9	Complement component C9
	P00751	CFB	Complement factor B
	P08603	CFH	Complement factor H
	Q92989	CLP1	Polyribonucleotide 5’-hydroxyl-kinase Clp1
	P32456	GBP2	Guanylate-binding protein 2
	P01876	IGHA1	Immunoglobulin heavy constant alpha 1
	P01877	IGHA2	Immunoglobulin heavy constant alpha 2
	P01857	IGHG1	Immunoglobulin heavy constant gamma 1
	P01859	IGHG2	Immunoglobulin heavy constant gamma 2
	P01860	IGHG3	Immunoglobulin heavy constant gamma 3
	P01861	IGHG4	Immunoglobulin heavy constant gamma 4
	P01871	IGHM	Immunoglobulin heavy constant mu
	P01834	IGKC	Immunoglobulin kappa constant
	P01602	IGKV1-5	Immunoglobulin kappa variable 1-5
	P04433	IGKV3D-11	Immunoglobulin kappa variable 3-11
	A0A0C4DH25	IGKV3D-20	Immunoglobulin kappa variable 3D-20
	P06312	IGKV4-1	Immunoglobulin kappa variable 4-1
	Q96PD5	PGLYRP2	N-acetylmuramoyl-L-alanine amidase
	Q9Y535	POLR3H	DNA-directed RNA polymerase III subunit RPC8
	P05155	SERPING1	Plasma protease C1 inhibitor
Glycosaminoglycan binding	Q96PD5	PGLYRP2	N-acetylmuramoyl-L-alanine amidase
	P01008	SERPINC1	Antithrombin-III
	P02749	APOH	Beta-2-glycoprotein 1
	O00622	CCN1	CCN family member 1
	P47914	RPL29	60S ribosomal protein L29
Glycosaminoglycan metabolic process	Q96PD5	PGLYRP2	N-acetylmuramoyl-L-alanine amidase
	P19823	ITIH2	Inter-alpha-trypsin inhibitor heavy chain H2
	P19827	ITIH1	Inter-alpha-trypsin inhibitor heavy chain H1

## Data Availability

The mass spectrometry proteomics data have been deposited to the ProteomeXchange Consortium via the PRIDE [68] partner repository with the dataset identifier PXD046125. For the raw data, T0 corresponds to H-NAIVE while TP corresponds to H-FCH.

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
