# Peer review of "Human Serum, Following Absorption of Fish Cartilage Hydrolysate, Promotes Dermal Fibroblast Healing through Anti-Inflammatory and Immunomodulatory Proteins"

_biomedicines, 2024, doi:10.3390/biomedicines12092132_

Round 1

Reviewer 1 Report

Comments and Suggestions for Authors

The study "Human serum enriched following fish cartilage hydrolysate absorption enhances the healing of primary human dermal fibroblasts through the expression of proteins with anti-inflammatory and immunomodulatory properties is interesting and provides novel data regarding the  beneficial impact of fish cartilage hydrolysate on human skin functionality by supporting wound healing.

However, there are certain observations that need interpretation by the authors before agreeing to publication:

Title

The title is too extensive, it should be shortened.

Abstract 

The abstract should have a background statement that emphasized the importance for conducting this study. The background part is long, however it does not provide clear reason for conducting this research.

Introduction

Line 43-45 Sentences are too simple and should be paraphrased to achieve the higher quality and scientific level.

The introduction contains a lot of information that is scattered and not well organized and logically connected. The authors successively state the results of previous studies and instead should highlight the summary and the crucial data of all studies and emphasize their importance.

Please provide a clear hypothesis to be tested in the study in this section.

The novelty is not highlighted, the authors only provided data from previous studies and it’s not clear what is the innovation of their research.

Material and methods

1. What were the inclusion and exclusion criteria for introduction to study protocol?

2. What time after receiving 12 g FCH has passed to blood sampling?

3. Provide the explanation for including only 10 volunteers in the study. The number is too low to be able to provide any scientific relevant information.

Discussion

Is there any data about the safety of FCH ingestion?

What would be the advantage of this approach compared to available treatment protocols for wound closure?

‘’Thus, the potential beneficial effects of FCH could be evaluated on skin explant models of AD or psoriasis’’ . It is not clear why the authors suggest the implementation of FCH for AD or psoriasis only when they confirmed wound healing potential as well.

Are parameters of inflammation studied in this manuscript relevant to provide conclusion that FCH can be used in AD or psoriasis? Can the experimental model used in this research mimic human skin disorders?

Conclusion

Provide future perspectives.

Comments on the Quality of English Language

English should be improved.

Reviewer 2 Report

Comments and Suggestions for Authors

The manuscript entitled "Human serum enriched following fish cartilage hydrolysate absorption enhances the healing of primary human dermal fibroblasts through the expression of proteins with anti-inflammatory and immunomodulatory properties" was well-written and organized. The authors considered FCH for enrichment of human serum and the results obtained revealed that  FCH’s beneficial impact on human skin functionality by supporting WH. minor consideration should be addressed for better presentation.

1. The introduction section need to be reconsidered. Some similar studies should be provided for comparison. 

2. The keywords section should be amended.

3. The conclusion section needs to be reconsidered.

4. The abbreviations should be rechecked. 

Comments on the Quality of English Language

 Minor editing of English language required.

Reviewer 3 Report

Comments and Suggestions for Authors

1.       The title of the paper is too long; it is recommended that the author shorten it.

2.       It is suggested to add a scale bar to Figure 2A.

3.       What does 0.06 in Figure 2B represent?

4.       It is recommended to enlarge the font in Figure 3.

5.       It is suggested that the author further elaborate on related works in the introduction that are pertinent to this study, such as Journal of Functional Biomaterials 14 (11), 553ï¼›Journal of Controlled Release 354, 821-834ï¼›Journal of Controlled Release 369, 591-603.

Comments on the Quality of English Language

Minor editing of English language required
